# Inhibitory Activity of *Saussurea costus* Extract against Bacteria, Candida, Herpes, and SARS-CoV-2

**DOI:** 10.3390/plants12030460

**Published:** 2023-01-19

**Authors:** Hajo Idriss, Babeker Siddig, Pamela González-Maldonado, H. M. Elkhair, Abbas I. Alakhras, Emad M. Abdallah, Amin O. Elzupir, Pablo H. Sotelo

**Affiliations:** 1Deanship of Scientific Research, Imam Mohammad Ibn Saud Islamic University (IMSIU), P.O. Box 5701, Riyadh 11432, Saudi Arabia; 2Department of Physics, College of Science, Imam Mohammad Ibn Saud Islamic University (IMSIU), Riyadh 11623, Saudi Arabia; 3Alawia Imam Institute for Pharmaceutical Research and Development, University of Medical Science and Technology, Khartoum 11115, Sudan; 4Savola Edible Oils, Khartoum 11115, Sudan; 5Biotechnology Department, Facultad de Ciencias Químicas, Universidad Nacional de Asunción, San Lorenzo 111421, Paraguay; 6Department of Chemistry, College of Science, Imam Mohammad Ibn Saud Islamic University (IMSIU), P.O. Box 90950, Riyadh 11623, Saudi Arabia; 7Department of Science Laboratories, College of Science and Arts, Qassim University, Ar Rass 51921, Saudi Arabia

**Keywords:** phytochemicals, *Candida albicans*, antibacterial, SARS-CoV-2, GC-MS, *Saussurea costus*, HSV-1

## Abstract

Medicinal herbs have long been utilized to treat various diseases or to relieve the symptoms of some ailments for extended periods. The present investigation demonstrates the phytochemical profile, molecular docking, anti-*Candida* activity, and anti-viral activity of the *Saussurea costus* acetic acid extract. GC-MS analysis of the extract revealed the presence of 69 chemical compounds. The chemical compounds were alkaloids (4%), terpenoids (79%), phenolic compounds (4%), hydrocarbons (7%), and sterols (6%). Molecular docking was used to study the inhibitory activity of 69 identified compounds against SARS-CoV-2. In total, 12 out of 69 compounds were found to have active properties exhibiting SARS-CoV-2 inhibition. The binding scores of these molecules were significantly low, ranging from −7.8 to −5.6 kcal/mol. The interaction of oxatricyclo [20.8.0.0(7,16)] triaconta-1(22),7(16),9,13,23,29-hexaene with the active site is more efficient. Furthermore, the extract exhibited significant antimicrobial activity (in vitro) against *Candida albicans*, which was the most susceptible microorganism, followed by *Bacillus cereus*, *Salmonella enterica*, *Staphylococcus aureus*, *Escherichia coli*, and *Pseudomonas aeruginosa*, respectively. On the other hand, its antiviral activity was evaluated against HSV-1 and SARS-CoV-2, and the results showed a significant positive influence against HSV-1 (EC_50_ = 82.6 g/mL; CC_50_ = 162.9 g/mL; selectivity index = 1.9). In spite of this, no impact could be observed in terms of inhibiting the entry of SARS-CoV-2 in vitro.

## 1. Introduction

Since antiquity, what we now refer to as folk, traditional, or alternative medicine has been the major source of remedies, which were mostly reliant on medicinal plants. On the other hand, modern medicine is now confronting a problem as a result of its inability to prevent the onset of disease. As a result, there is a pressing need to return to Mother Nature and its wealth of natural medicines [1,2].

In the scientific literature, numerous studies on the bioactive properties of medicinal plants have been published. For example, there are claims that *Allium sativum*, *Trigonella foenum-graecum*, *Ferula assa-foetida*, *Carthamus tinctorius*, and *Mangifera indica* are all plants with antidiabetic activity [3]. *Pistacia lentiscus*, *Diospyros abyssinica*, *Sargentodoxa cuneata*, *Acacia auriculiformis*, *Ficus microcarpa*, *Salvia officinalis,* and the *Lamiaceae* species contain potent antioxidants [4]. *Aloe vera, Curcuma longa, Emblica officinalis, Stevia rebaudiana*, and *Gymnema sylvestre* are suggested for possible anticancer activity [5]. *Stephania glabra, Woodfordia fruticosa, Betula utilis, Nelumbo nucifera*, and *Calotropis gigantean* have been identified as effective antimicrobial plants (in vitro) [6]. Recently, some medicinal plants have been reported to possibly possess anti-SARS-CoV-2 activity (in vitro), such as *Lycoris radiate, Artemisia annua, Pyrrosia lingua, Nigella sativa*, and *Houttuynia cordata* [7]. It is important to mention that clinical trials are required to demonstrate all these possible activities, particularly regarding SARS-CoV-2, for which no effective drug based on natural products has been approved yet.

*Saussurea costus* (Falc.) Lipschitz, also known as *Saussurea lappa* C.B. Clarke, is a well-known and valuable medicinal plant that is used in many indigenous medical systems to treat asthma, inflammatory diseases, ulcers, and stomach disorders, among other conditions [8]. Moreover, some studies have reported that *Saussurea costus* exhibits antiviral activity [9,10].

In January 2020, the World Health Organization (WHO) declared the outbreak of the COVID-19 pandemic due to the global spread of the disease and the novel emergency alarm raised by the WHO [11]. Electronic health records with easily accessible characteristics in combination with a COVID-19 mortality risk prediction model may allow rapid and precise risk categorization of COVID-19 patients upon admission.[12]. Coronaviruses are a broad group of respiratory viruses that are responsible for various disorders, including the common cold [13]. The new virus is unique due to its complex envelope protein, which resulted from mutations [14]. A coronavirus is an enveloped RNA virus that is covered by the spike protein. This protein assists the virus in entering the host and determining the properties of the host [15]. The protein has two parts: a membrane protein and an envelope protein, both of which are considered to have a role in the progression of the sickness [16]. In addition, hemagglutinin esterase is a glycoprotein that is present in bovine coronaviruses and is attached to their envelopes [17].

According to WHO statistics, the number of infected cases of COVID-19 has reached more than four hundred million globally, with a 1.3% death rate and 86.4% of patients treated [18]. The severe conditions have motivated global researchers to organize research teams to cover broad research areas in different fields, including drug discovery [16], mathematical modeling and data management, sociopolitical analysis, and education. Furthermore, various types of medicinal plants and herbs have been utilized for their possible antiviral activity. Several common ailments, including malaria, cholera, and asthma, are treated using medicinal herbs [19]. Although medicinal herbs are adjuvant treatments, the protocols to treat these diseases are still based on synthetic medicines. *Heteromorpha* spp. medicinal plants have demonstrated antiviral activity against HCV-229E (in vitro) [20]. *Eleutherococcus senticosus* extract has proven to be an RNA viral inhibitor [21]. A polar water-soluble natural compound extracted from *Sannicolas europium* was found to inhibit an RNA virus [22]. Medicinal natural compounds extracted from *Rosa nutkana* and *Amelanchier alnifolia* exhibited vigorous activity towards enteric coronavirus [23].

On the other hand, the crisis of drug-resistant microbes is growing globally. It has been reported that approximately 13 million deaths per year are attributed to microbial infections (other than viral infections), despite efforts to prevent and manage microbial pathogens that have been ongoing for more than a century and which have been largely successful with the aid of antibiotics. However, these antibiotics are currently failing to control newly emerging and re-emerging bacterial contagious diseases [24]. Bacteria, fungi, and parasites develop the ability to avoid the effects of antimicrobial drugs, survive, and, in some circumstances, grow extremely virulent. The process underpinning the development of drug-sensitive and drug-resistant microorganisms is a phenomenon of great complexity. Implementing an antimicrobial stewardship program is vital to controlling this growing phenomenon, and more efforts are needed to develop new antimicrobial medications [25].

Many medicinal herbs such as *Allium sativum* [26], *Azadirachta indica* [27], *Tinospora Cordifolia* [28], *Syzygium aromaticum* [29], *Panax quinquefolius* [30], *Piper nigrum* [31], *Withania somnifera* [32], *Curcuma longa* [33], *Sambucus nigra* [34],and *Tinospora Cordifolia* [35,36] have been approved as sources of minerals, steroids, and alkaloids and have been suggested for possible antiviral activities (in vitro). *Saussurea Costus*, a medicinal plant, is grown in vast regions worldwide, including India, Pakistan, and some parts of the Himalayas [37].The taxonomic details of *Saussurea costus* are as follows: its *Plantae* phylum is *Trichophyte*, its class is *Magnoliopsidas*, its order is *Astral*, and it belongs to the *Asteraceae* family [38]. *Saussurea costus*, in terms of morphology, is a 1–2 m tall upright herbaceous plant with a strong and hairy seedhead. Leaves are narrow, rough, glabrous, and auriculate and have irregularly formed teeth in general. At 0.50 to 1.25 m in length, the bottom leaves are somewhat large and feature long petioles with wings. Upper leaves are tiny and virtually sessile, with two small lobes at the base that almost wrap around the stem. The flowers are stemless and range in color from bluish-purple to nearly black. Their width varies from 2.4 to 3.9 cm, and they are spherical. The roots are dark brown or gray and reach a length of 40 cm [38]. The roots of *Saussurea costus* have been used for treating dysentery, rheumatism, stomachache, ulcer, and toothache [39]. The roots of *Saussurea costus* are widely used in Islamic culture and are famous in Arab regions, although it is not cultivated there. Moreover, traditional healers claim that it can control COVID-19 [40]. Studies on the potential anti-COVID-19 activity and the antimicrobial potential of *Saussurea costus* on specific fractions are scarce. Therefore, the main objective of the current study was to determine the phytochemical compounds in the acetic extract of *Saussurea costus* and investigate the possible in vitro antibacterial and anti-*Candida*, anti-herpes, and anti-SARS-CoV-2 potential, as well as its molecular docking to understand the possible interactions of these bioactive molecules at the atomic level.

## 2. Results and Discussion

### 2.1. Analysis of the Organic Chemical Compounds in Saussurea costus

Appendix A displays the GC-MS chromatograms of *Saussurea costus* extracted using acetic acid. We identified 109 peaks from the chromatograms that correspond to chemical substances by comparing their peak retention times, peak area percentages, height percentages, and mass spectral fragmentation features to those of known compounds in the National Institute of Standards and Technology (NIST) library [41]. As indicated in Appendix A, phytochemical studies of the acetic acid extraction of *Saussurea costus* roots indicated the existence of 69 compounds. The percentages of the phytochemicals were as follows: alkaloids, 5%; terpenoids, 79%; phenolic compounds, 4%; hydrocarbons, 7%; and sterols, 6%. According to these findings, terpenoids accounted for the highest percentage of organic chemical compounds (78%), whereas alkaloids and phenolic compounds accounted for the lowest percentages (4%) of chemical compounds, as depicted in Figure 1. Terpenoids included 0.02% (4-Terpinenyl acetate) and 13.23% Vanillosmin. Phenolic compounds ranged between 0.01 (Phenol, 2-methoxy-4-(2-propenyl)-, acetate) and 0.06% (Phenol, 2,6-dimethoxy). Alkaloids ranged from 0.51% (Piperine) to 0.04% (Di(1,2,5-oxadiazolo)[3,4-b:3,4-E]pyrazine, 4,8-diacetyl). Hydrocarbons varied from 0.05% (1-Heptadecene) to 0.37 (9-Methyl-10,12-hexadecadien-1-ol acetate). Sterols ranged from 0.26% (Ergost-5-en-3-ol, (3.beta.)) to 3.51% (9,19-Cycloergost-24(28)-en-3-ol, 4,14-dimethyl-, acetate (3.beta.,4.alpha.,5.alpha)). Through a comparison of the chemical profile obtained in the current study with other published data, we found a difference in the nature and number of compounds and their concentrations [42,43]. Various studies have shown that chemical compound concentrations vary depending on different factors, which include the solvent used in plant extracts, the area of plants grown, and the quantity of the plant used [44]. The active constituents of this well-known medicinal plant are mostly terpenes, with various levels of flavonoids, anthraquinones, alkaloids, tannins, and inulin, as described in earlier research [45]. Several laboratory experiments and animal model studies have indicated that *Saussurea costus* possesses anti-inflammatory, antitrypanosomal, and anti-malignant tumor effects, confirming its extended use in medicine [46,47]. Costsunolide, dehydrocostus lactone, and cynaropicrin are some of the phytochemicals found in this plant that have shown promise in producing bioactive molecules [48]. Because *Saussurea costus* has displayed solid anticancer, anti-inflammatory, antibacterial, antifungal, and antiviral action, it will be possible to improve its use as a medication soon. According to several studies, sterol compounds act as cancer inhibitors, anti-inflammatory medications, immunomodulatory agents, and antiviral agents [49,50]. Various investigations have shown that terpenoids have a high proclivity for acting as SARS-CoV-2 blockers [51]. Based on a new study integrating quantum chemistry, molecular docking, and dynamic dynamics, phenolic chemicals have potential therapeutic implications for SARS-CoV-2. As a result, the mixture of active components present in each plant may be more effective than a single separated molecule in terms of treatment efficacy [52]. As reported in some investigations, researchers have estimated that the quinoline-2-carboxylic acids identified in *Ephedra sinica* might be employed as COVID-19 medicinal agents [53].

### 2.2. Molecular Docking

This study had a particular focus on the important residue GLU 166 and the catalytic dyad CYS 145, as well as HIS 41 of the main protease (M^pro^). All of the 69 compounds identified were tested against M^pro^. Among these, 12 were docked to the active site of the M^pro^ (Figure 2). As shown in Table 1, they had low energy scores and interacted with the active sites in the pocket of the M^pro^ enzyme. The binding scores of these molecules were significantly low, ranging from −7.8 to −5.6 kcal/mol. The interaction of oxatricyclo [20.8.0.0(7,16)]triaconta-1(22),7(16),9,13,23,29-hexaene with the active site has generally been shown to be more efficient. As sterols interact with the active site, they can be promising candidates, particularly oxygenous tetrahydroxy sterol, which has a high tendency to form hydrogen bonds. Schiff base ligands have been shown to inhibit M^pro^ in previous studies [54] as well as caffeine [54,55], nethylxanthines [56], natural product isolates [57], glycyrrhizin [58], ML188 [59], pyrimidonic and pyridonic pharmaceuticals [60,61], kaempferol [62], fungal natural products [63], marine natural compounds [64], Ceftazidime [65], hepatitis C virus protease drugs [66], and other inhibitors [67,68,69,70,71,72]. In our molecular docking study, these natural compounds were identified as potential M^pro^ inhibitors.

### 2.3. Antimicrobial Activity

In the present study, the antimicrobial activity of *Saussurea costus* extracts (aqueous and acetic acid extracts) was analyzed qualitatively and quantitatively. The results of the disc-diffusion test are represented in Table 2. According to the results, the acetic acid extract revealed significant antimicrobial activity against all tested microorganisms. The most susceptible microorganism was *Candida albicans* with 38.5 ± 1.5 mm, followed by *Bacillus cereus* with 25.0 ± 2.0 mm, *Salmonella enterica* with 18.0 ± 0.0 mm, *Staphylococcus aureus* with 16.5 ± 0.5 mm, *Escherichia coli* with 14.0 ± 1.0 mm, and *Pseudomonas aeruginosa* with 13.5 ± 0.5 mm. Our results are in agreement with previous published studies which showed that the methanol and ethanol extracts of *Saussurea lappa* showed significant antibacterial and antifungal activity [73]. Essential oils of *Saussurea costus* roots were also cited to have high antimicrobial efficacy [74]. Our results revealed noticeable anti-*Candida* activity higher than the reference antibiotic (clotrimazole, 5 mg/mL), which was in agreement with previously published reports showing that *Candida albicans* was highly susceptible to the semi-polar and non-polar extracts [75]. The antimicrobial activity can be attributed to some phytochemical molecules in the acetic acid extract.

The results of MIC, MBC, MFC, MBC/MIC, and MFC/MIC analyses are represented in Table 3. These results support those of the disc-diffusion test, which showed that the MIC, which was the lowest concentration of *Saussurea costus* acetic acid root extract that inhibited the visible growth of microorganisms after overnight incubation, was 25 mg/mL for the bacterial stains and 6.25 mg/mL for *Candida albicans*. The results suggest that this yeast is highly susceptible to *Saussurea costus* and indicate that this plant contains significant organic active ingredients for treating pathogenic *Candida* spp. Diseases caused by *Candida* have become much more common around the world, and in some patient groups, the death rate is over 70% [76]. *Candida* spp. causes cutaneous, gastrointestinal, and vaginal infections, as well as severe vaginitis, endophthalmitis, and candidiasis for medically compromised patients [77]. Moreover, previous research showed that dehydro-costus lactone is the major anti-candida component of this plant [54]. The values of MBC (for bacteria) and MFC (for the fungus), which show the minimum concentrations of *Saussurea costus* acetic acid extract that killed 99.9% of the examined microorganisms, were 50 mg/mL for all bacteria except for 25 mg/mL for *Staphylococcus aureus* and 12.5 mg/mL for *Candida albicans* (MFC). Additionally, the MBC/MIC was less than four, leading us to conclude that the extract had a bactericidal effect on the bacterial strains, and the MFC/MIC ratio was also less than four, which means that it demonstrated fungicidal activity against *Candida albicans*. When an extract’s MBC/MIC ratio is less than four, the extract is regarded as bactericidal, and if the ratio is more than or equal to four, the extract is deemed bacteriostatic [78]. Furthermore, a compound is classified as fungistatic when the MFC/MIC ratio is greater than or equal to four and as fungicidal when the MFC/MIC ratio is less than four [79]. Therefore, we concluded that *Saussurea costus* acetic acid extract has bactericidal and fungicidal properties.

### 2.4. Antiviral Activity by Saussurea costus Acetic Extract

The inhibitory activity of the *S. costus* acetic extract against SARS-CoV2 was evaluated using the pseudovirus method. This system allows the investigation of inhibitors of virus entry into the cell [80]. To evaluate the effect of the extract on variants of concern (VOCs) of SARS-CoV2, the antiviral activity was observed using D614G, Gamma, and Delta variants of the pseudotyped virus. No antiviral effect of the acetic extract from *Saussurea costus* was observed with any SAR-CoV2 variants (Figure 3), and no differences between different variants were observed using one-way ANOVA. These results are in agreement with previous results indicating that the aqueous extract of *S. costus* lacks entry inhibitory activity against SARS-CoV2 [54]. Furthermore, the antiviral activity of *S. costus* acetic extract against HSV-1, another enveloped virus, was evaluated. For this purpose, serial dilutions of the extract were added post-virus-adsorption, and the amount of virus produced was evaluated by qPCR. The *S. costus* acetic extract showed antiviral activity with an effective concentration 50 (EC50) of 82.6 µg/mL, a cytotoxic concentration (CC50) of 162.9 µg/mL, and a selectivity index of 1.9 (Figure 4 and Figure 5).

A time-of-infection study was performed to identify the extract’s effect on the viral cycle. The extract was added at different times during infection. In the pre-infection condition, the acetic extract was added 2 h before the addition of the virus, then it was removed, and the cells were infected. In the adsorption condition, the virus was added at the same time as the extract and incubated for one hour and then replaced with the medium. Under the post-entry condition, the extract was added after the virus entered the cell. The *Saussurea costus* acetic extract showed pre-infection and post-entry antiviral activities, indicating that its compounds act by different antiviral mechanisms.

In a previous study, the antiviral activity of the aqueous extract of *S. costus* (EC50 = 1.35 mg/mL, CC50 = 4.92 mg/mL, and SI = 3.6) was reported. In the present work, a similar behavior was observed with the *S. costus* acetic extract (EC50 = 82.6 mg/mL, CC50 = 162.9 mg/mL, and SI = 1.9). However, a difference in the mechanisms of action of the extracts was observed. The aqueous extract only showed post-entry antiviral activity, whereas the acetic extract also inhibited the virus in the pre-infection condition. This result suggests that the acetic extract also induced an antiviral state in cells.

## 3. Materials and Methods

### 3.1. Chemicals

All chemicals used (acetic acid, dimethyl sulfoxide, ethanol, anhydrous sodium sulfite) were of analytical grade, were used as received without any further purification.

### 3.2. Sample Preparation and Extraction

Roots of *Saussurea costus* of high quality were purchased from “the attar shop”, which is a trusted herbal store in Riyadh, Saudi Arabia, with a license from the government to sell herbs; the source of *Saussurea costus* is Kashmir in the northwestern region of India. The root of *Saussurea costus* was air-dried at room temperature for many days. The dried roots were ground using an electric grinder. Dried and powdered samples of *Saussurea costus* acetic acid solvent (99.8%) extract (Scharlau Sentmenat, Barcelona, Spain) underwent filtration, and the filtrate was allowed to dry on a glass Petri dish at room temperature. Fifty grams of dried *Saussurea* roots were extracted in 150 milliliters of acetic acid at room temperature for four days with an Orbital Shaker (BioSan PSU-20i, Riga Latvia).

### 3.3. Gas Chromatography–Mass Spectrometry (GC/MS) Analysis

A GM/MS system from Shimadzu Kyoto Japan, with serial number 020525101565SA and a capillary column (Rtx-5MS 30 m 0.25 mm × 0.25 mm), was employed for the analysis of samples. The solution of the samples was passed through a 0.45 mm syringe filter and into a 1.5 mL GC-MS vial, where it was prepared for injection. The specimen was injected utilizing split mode with helium serving as the carrier gas, which had a 1.61 mL/min flow rate and moved inside the injector. According to the manufacturer’s instructions, the temperature program started at 50 °C at a rate of 10 °C per minute and ended at 300 °C with a hold duration of 10 min. The injection port temperature was 300 degrees Celsius; the ion source was 200 °C, and the interface was 250 °C. The sample required 35 min for the analysis in scan mode in the m/z 40–500 mass-to-charge ratio range, with a 40-min run time. (GC-MS analysis conditions are shown in Appendix A). The sample’s contents were identified by comparing the retention index and mass fragmentation patterns of the sample to those in the National Institute of Standards and Technology library (NIST). The relative quantities of the different components were determined without correction factors based on the peak area of the GC (FID response).

### 3.4. Molecular Docking

As described in PubChem, Canonical SMILES was used to prepare the 3D models for molecular docking, along with Open Babel. Using the Protein Data Bank database, we downloaded the M^pro^ of SARS-CoV-2’s crystal structure (PDB ID: 6Y2E). Then, water residues were removed, and their energy was minimized using the Molecular Modeling Toolkit plugin UCSF Chimera [60,81,82,83]. Molecular docking was accomplished using AutoDock Vina with a grid box of (−16.5 × −24.0 × 16.5) Å, and it was centered at (35.0, 65.0, 65.0) Å. UCSF Chimera was used for the visualization of images.

### 3.5. Microorganisms

In this investigation, six American type culture collection (ATCC) microbial isolates were utilized, including Gram-positive bacteria (*Staphylococcus aureus* ATCC BAA 1026 and *Bacillus cereus* ATCC 10876), Gram-negative bacteria (*Pseudomonas aeruginosa* ATCC 10145, *Salmonella enterica* ATCC 14028, and *Escherichia coli* ATCC 9637), and a yeast (*Candida albicans* ATCC 10231). All of the isolated microorganisms were provided by the Department of Laboratory Sciences at Al-Rass, Qassim University in Saudi Arabia.

### 3.6. Disc-Diffusion Test

The disc-diffusion test was used to evaluate the initial antimicrobial activities of the extract against the selected microbial strains, using the previously reported procedure with minor modifications [84]. Briefly, the dried acetic acid extract of *Saussurea costus* roots was re-constituted by dissolving 500 mg of the dry extract in 10% dimethyl sulfoxide (DMSO) and mixed well using a shaker for up to 1 h. According to our pre-experimental evaluation and the literature, 10% DMSO has no inhibitory effect on microbial growth [85,86,87]. A fresh microbial suspension adjusted to 0.5 McFarland turbidity (10^8^ CFU/mL) (CFU: colony-forming units) was streaked over sterile plates containing Mueller–Hinton Agar for bacteria or Sabouraud Dextrose Agar for the yeast (microbial media were obtained from Oxoid, UK). Under aseptic conditions, sterile paper discs (6 mm) were carefully saturated with 10 µL of the reconstituted extract (100 mg/mL) using an Eppendorf pipette. Saturated papers were immediately loaded over the streaked plates and left to settle for up to 15 min, and then the seeded plates were incubated upside-down at about 37 °C for 24 hours for bacteria and for 48 hours for *Candida albicans*. A disc containing 10 µL of 10% DMSO served as a negative control, whereas a disc containing chloramphenicol (2.5 mg/mL) for bacteria and clotrimazole (5 mg/mL) for yeast served as a reference drug (positive control). The inhibitory diameter was then measured in millimeters (disc included) and reported as the mean ± standard deviation for two trials.

### 3.7. Minimum Inhibitory Concentration MIC Assay

The minimum inhibitory concentration (MIC) values were determined using the micro-well dilution method for the microorganisms determined to be sensitive to the extract in the disc diffusion experiment [88]. The microorganism inoculum was taken from stock culture (in slant tubes containing general-purpose nutrient agar, Oxoid, UK) for bacteria or Sabouraud dextrose agar (Oxoid, UK) for the yeast and inoculated in nutrient broth (general-purpose growth medium, Oxoid, UK for bacteria or Sabouraud dextrose broth for the yeast and incubated for up to 12 h and up to 24 h for the yeast to reach the exponential phase. Then, microbiological suspensions were adjusted to a turbidity standard of 0.5 McFarland units. The dry crude extract was diluted in 10% dimethyl sulfoxide (DMSO) to obtain a concentration of 100 mg/mL, and then two-fold dilutions were prepared in test tubes containing nutrient broth over a concentration range of 1.5 to 100 mg/mL. In order to prepare the 96-well plates, 100 µL of the previously prepared serial dilutions were transferred into each of seven successive wells. Then, 95 µL of nutrient broth for bacteria or Sabouraud dextrose broth for the yeast and 5 µL of microbial inoculum were added to each well, bringing the total amount to 200 µL. As a negative control in another well series, nutrient broth or Sabouraud dextrose broth with sterile normal saline instead of extract and 5 mL of the inoculum were applied to each well. Additionally, successive dilutions of the standard antimicrobial drugs (chloramphenicol or clotrimazole) were used as positive controls. The 96-well plates were covered using a sterile sealant. The contents of each well were shaken on a plate shaker at 300 rpm for 20 s, before being incubated for 24 h for bacteria of 48 h for yeast at the specified temperatures. Microbial growth was estimated by measuring the absorbance at 595 nm using an iMark Absorbance microplate reader (BIO-RAD Inc., Hercules, California, USA). In this investigation, the extract was screened twice against each organism. The MIC is the lowest concentration of a substance that inhibits the growth of microorganisms.

### 3.8. Minimum Bactericidal Concentration or Minimum Fungicidal Concentration MBC or MFC Assay

The minimum bactericidal concentration (MBC) or minimum fungicidal concentration (MFC) can be defined as the lowest concentration at which an antimicrobial agent will eradicate a certain microorganism. The agar diffusion test was used here with minor modifications [89]. After the MIC test was completed, the MBC or MFC test was conducted. In a nutshell, 50 µL from each MIC tube was taken using an Eppendorf pipette and spotted onto nutritional agar plates for bacteria or Sabouraud dextrose agar for the yeast. These plates were then incubated overnight at 30 °C –35 °C. MBC was defined as the lowest MIC that showed no detectable growth. Additionally, MBC/MIC values for bacteria and MBC/MFC values for *Candida albicans* were calculated to classify the extract as bactericidal/fungicidal (4 or less) or bacteriostatic/fungistatic (more than 4).

### 3.9. Virological Assessments

#### 3.9.1. Cytotoxicity Assays

Cytotoxicity assays were performed as previously described [90,91]. A stock solution of 100 mg/mL of the re-constituted acetic acid extract of *Saussurea costus* in DMSO was made. Human ACE2 Stable Cell Line and HEK293T: is a popular derivative of the original HEK293 parent cell line (HEK293T-ACE2) or Vero cells were cultured at 1 × 10^4^ cells/well in a 96-well plate in the presence of different concentrations of the *S. costus* extract in DMEM 10% FBS (Foetal Bovine Serum) for HEK293T-ACE2 cells and in DMEM 2% FBS in the case of Vero cells. DMSO-treated cells were used as a control and normalizer. After 48 h, resazurin was added to a final concentration of 0.0015% per well. Absorbance was measured on the Multiskan TM GO (Thermo Scientific, USA) at 570 and 630 nm.

#### 3.9.2. HIV-1-Based SARS-CoV-2 Pseudotyped Particles

The plasmids used were as follows: pNL4.3-ΔEnv-FLuc, spike-G614-Δ19 (10.1126/sciadv.abe6855), pcDNA3.3_CoV2_P1 (Addgene plasmid #170450; http://n2t.net/addgene:170450; RRID: Addgene_170450) [92], and pcDNA3.3-SARS2-B.1.617.2 (Addgene plasmid # 172320; http://n2t.net/addgene:172320; RRID: Addgene_172320) [93]. The plasmids pcDNA3.3_CoV2_P1 and pcDNA3.3-SARS2-B.1.617.2 were a gift from David Nemazee.

Pseudotyped viral particles were generated by transfecting the plasmids (20 µg, molar ratio 3:2) pNL4.3-ΔEnv-FLuc and spike-G614-Δ19 or pcDNA3.3_CoV2_P1 (spike-GammaΔ18) or pcDNA3.3-SARS2-B.1.617.2 (spike-DeltaΔ18) with 2.5 M calcium chloride, and HEPES-buffered saline 2X (0.28 M NaCl, 0.05 M HEPES, 1.5 mM Na2HPO4, pH 7.05) [94]. The pseudotyped virus-containing supernatant was collected 48 h after transfection and centrifuged at 3500 rpm for 5 min at room temperature and stored at −80 °C. Stocks were titrated with HEK293T-ACE2, and firefly luciferase activity was measured 48 h later using the Dual-Luciferase Reporter Assay System kit (Promega, Madison, WI, USA) and Fluoroskan FL (Thermo Scientific). HEK293T (lacking the ACE2 receptor) cells transduced with the pseudotyped particles were used as a negative control.

#### 3.9.3. Inhibition of Severe Acute Respiratory Syndrome Coronavirus 2 (SARS-CoV-2) Entry Assay

The assay was performed in HEK293-ACE2 cells, as described previously [95]. Briefly, 1 × 10^4^ cells in suspension were added to each well of 96-well plates and infected in the presence of 62.5 µg/mL of the re-constituted acetic acid extract of *Saussurea costus* in DMEM 10% FBS. After 48 h, the firefly luciferase activity was measured using Dual-Luciferase Reporter Assay System kit (Promega, USA) and Fluoroskan FL (Thermo Scientific). HEK293T-ACE2 cells transduced with the pseudotyped virus without the extract were used as control untreated cells. The following formula was used to calculate the percentage of inhibition: 100 − (RLUs of treated cells/RLUs of control untreated cells) × 100.

#### 3.9.4. Antiviral Activity against Herpes Simplex Virus Type-1 (HSV-1) 

Antiviral activity assays were performed as previously described [96]. Briefly, Vero cells were seeded in 96-well plates at 1 × 10^4^ cells/well in DMEM 7.5% FBS. The next day, the cells were infected at MOI 1.5 in DMEM 2% FBS. After 1 h of virus adsorption, the medium was replaced, and dilutions of the extract in DMEM 2% FBS were added. Control-infected cells were incubated with DMSO. Forty-eight hours later, the virus in the supernatant was quantified via qPCR, using 5 µL of SYBR^®^ Green Master Mix buffer (Bio-Rad) 2X, 500 pM of each primer, and 1 µL of sample as a template. Amplifications were carried out on a StepOnePlus™ thermocycler (Applied Biosystems). A two-step program of 10 min at 95 °C, 35 cycles of 15 s at 95 °C, and 30 s at 60 °C, and a final melting curve was used. The viral genome copies (VGC) were calculated using a calibration curve.

#### 3.9.5. Time-of-Addition Assay of HSV-1

The time-of-addition assay was carried out under three different experimental conditions according to the time at which the extract was added. In the pre-infection condition, the extract was added two hours before viral adsorption. Subsequently, cells were washed with PBS, the virus was added, and the infection was performed as described previously. Under the adsorption conditions, the extract was added with the virus and was incubated for 1 h to 37 °C. After that, the viral inoculum was removed, cells were washed with PBS and replaced with DMEM 2% FBS, and the infection was performed as previously described. In the post-entry condition, the viral inoculum was added and incubated for 1 h at 37 °C and washed with PBS, and the extract in DMEM 2%FBS was added. Forty-eight hours later, the virus genome in the supernatant was quantitated via qPCR, and the antiviral activity was quantified as previously described [96].

### 3.10. Statistical Analysis

In this study, the data are shown as the mean standard error of the mean. The one-way ANOVA (analysis of variance) method and SPSS 14.0 (SPSS Inc., Chicago, IL, USA) were used to determine whether there were statistically significant differences between the microorganisms that were tested.

### 3.11. Research Limitations

Experiments carried out on *Saussurea costus* roots grown in India and those carried out on the same plant grown in other tropical and subtropical regions may show some variations in results. The antimicrobial experiments were repeated twice, and the mean was calculated.

## 4. Conclusions

In summary, GC-MS analytical screening of *Saussurea costus* extract using acetic acid was investigated. The survey outcomes resulted in the detection of 69 phytochemical substances in the acetic acid extract. The order of phytochemicals showed the following trend: terpenoids > hydrocarbons > sterols > alkaloids = phenolic compounds. The crude root extract of *Saussurea costus* obtained using acetic acid displayed significant antibacterial activity. The inhibitory activity of sterols and terpenoids against SARS-CoV-2 was studied using molecular docking. The examined sterols and terpenoids were found to be active. Furthermore, sterols generally exhibited a more efficient interaction with the active site. *Candida albicans* was the most vulnerable organism, followed by *Bacillus cereus*, *Salmonella enterica*, *Staphylococcus aureus*, *Escherichia coli*, and *Pseudomonas aeruginosa*, respectively. Evaluations of antiviral activity against HSV-1 and SARS-CoV2 revealed a considerable beneficial effect against HSV-1 (EC50 = 82.6 g/mL; CC50 = 162.9 g/mL; selectivity index = 1.9). However, no effect was detected in terms of the inhibition of SARS-CoV2 entry.

## Figures and Tables

**Figure 1 plants-12-00460-f001:**
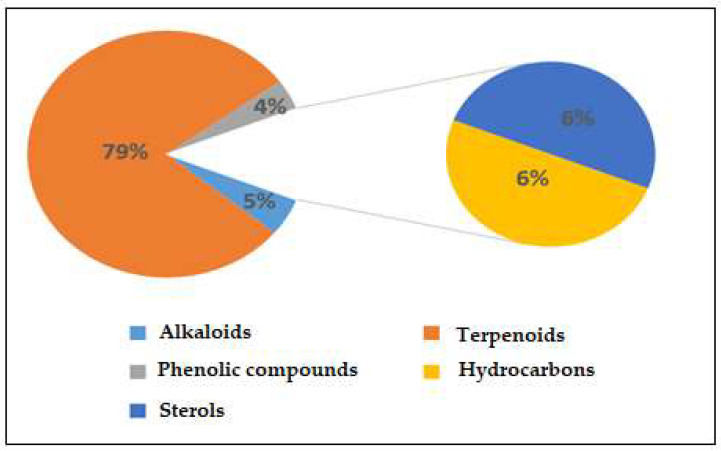
Percentages of chemical compounds extracted by acetic acid from *Saussurea costus*.

**Figure 2 plants-12-00460-f002:**
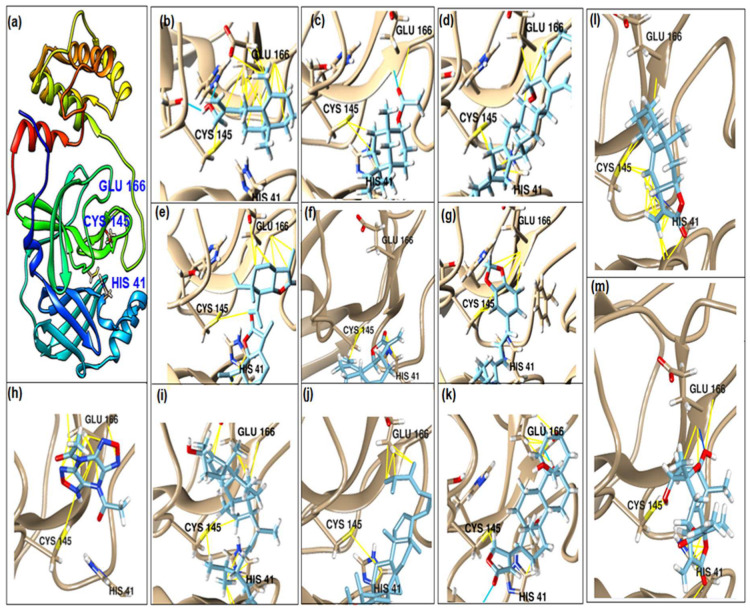
(**a**) The main protease of the SARS-CoV-2 crystal structure, with a particular focus on GLU 166, CYS 145, and HIS 41 active residues (PDB ID: 6Y2E). Amplified picture of docked compound (**a**–**m**) with the active site of M^pro^.

**Figure 3 plants-12-00460-f003:**
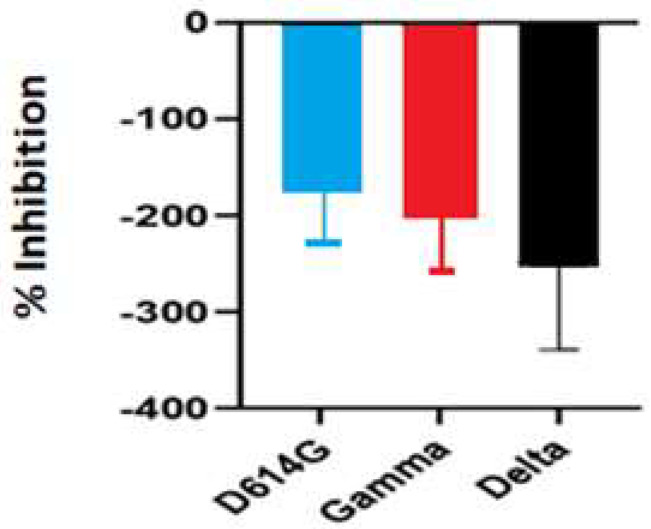
Antiviral activity of *Saussurea costus* acetic extract against SAR-CoV2. HEK-293T ACE2 cells were infected with the corresponding pseudotyped virus in the presence or absence of the extract. After 48 h, the luciferase activity was measured. The percentage of inhibition was determined as the ratio between treated and untreated cells.

**Figure 4 plants-12-00460-f004:**
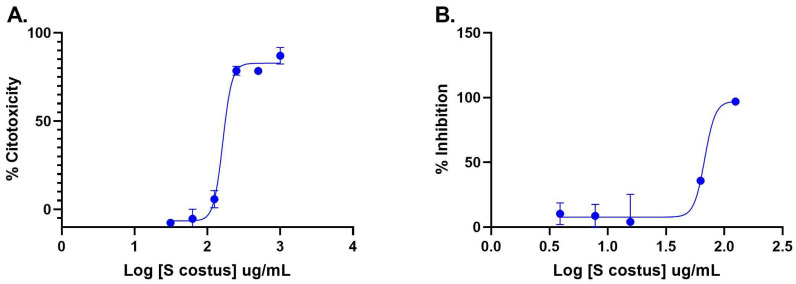
Anti-herpetic activity and cytotoxicity of *Saussurea costus* extract. (**A**) Vero cells were incubated with increasing concentrations of the acetic extract, and after 72 h, the cytotoxicity was measured as described in the Materials and Methods. (**B**) Vero cells were infected at MOI 1.5 with HSV-1 and incubated with increasing concentrations of the extract. After 48 h, the virus genome was quantitated in the supernatant via qPCR. The percentage of inhibition was determined as the ratio between treated and untreated infected cells. Data are expressed as mean +/− SD for *n* = 2.

**Figure 5 plants-12-00460-f005:**
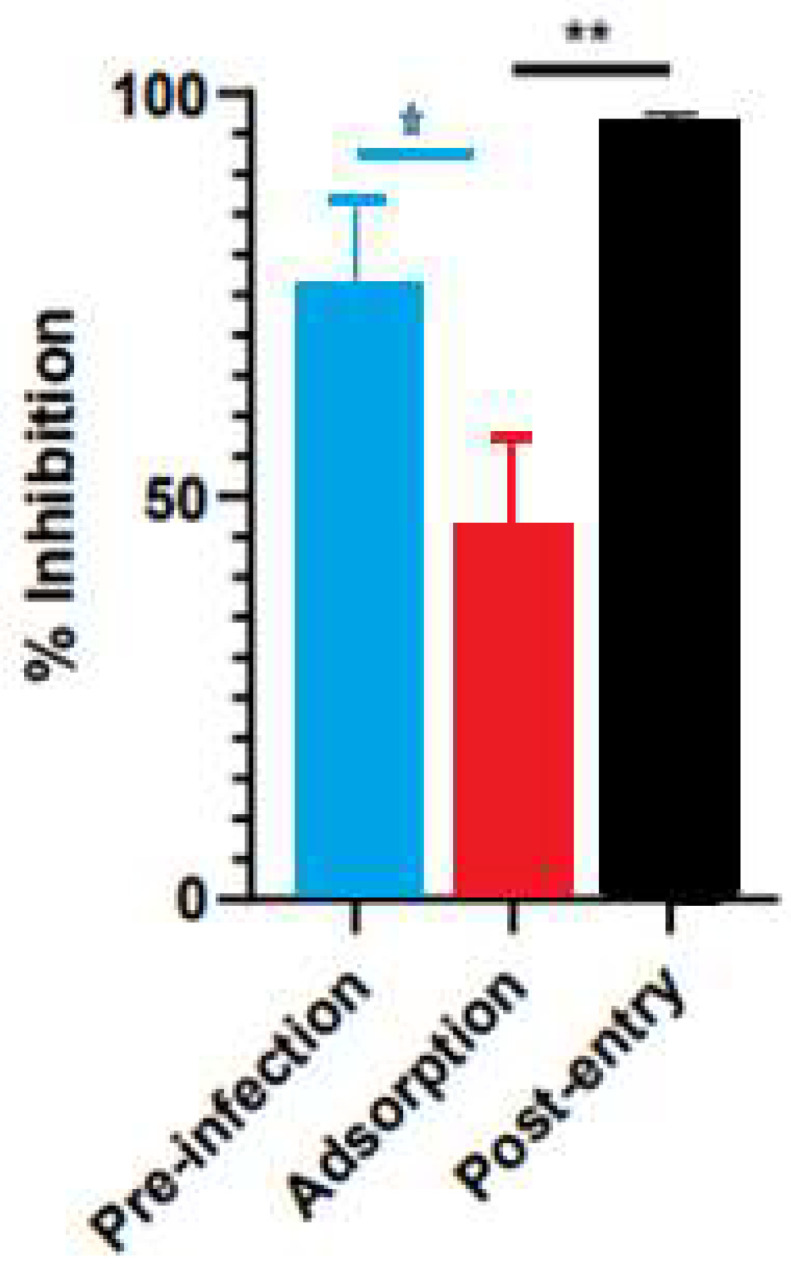
Time-of-addition assay results for *Saussurea costus* acetic extract. Vero cells were infected at MOI 1.5 and incubated with the extract at different times during the infection, as described in the Materials and Methods. After 48 h post-infection, the virus genome was quantitated in the supernatant via qPCR. The percentage of inhibition was determined as the ratio between treated and untreated infected cells. Data are expressed as mean +/− SD for n = 3. Statistical analysis was performed using one-way ANOVA, followed by multiple comparisons testing with significance indicated as * *p* < 0.05, ** *p* < 0.01.

**Table 1 plants-12-00460-t001:** The binding affinity of the active compounds docked with the active site of the main protease of SARS-CoV-2 with a particular focus on the GLU 166, HIS 41, and CYS 145 residues.

No.	Compound Name	Chemical Structure	Score (kcal/mol)	Root-Mean-Square Deviation of Atomic Positions RMSD Range	Hydrogen Bonds (Number of Bonds/Number of Conformations), (Distance ≤ 4 Å)	Van der Waals (Number of Bonds/Number of Conformations), (Distance Range Å)
b	2-(4a,8-Dimethyl-1,2,3,4,4a,5,6,7-octahydro-naphthalen-2-yl)-prop-2-en-1-ol	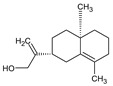	−5.6	27.83–29.77	HIS 163 (1/1), SER 144 (1/1)	GLU 166 (19/1), (2.52–3.87) CYS 145 (1/1), (4.01)
d	9beta-Acetoxy-3,5alpha,8-trimethyltricyclo[6.3.1.0(1,5)]dodec-3-ene	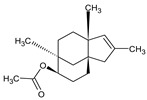	−5.6	33.48–36.10	GLU 166 (1/1)	GLU 166 (5/2), (2.01–3.80) HIS 41 (6/2), (3.11–3.80) CYS 145 (4/2), (3.76–3.95)
c	3-Oxatricyclo[20.8.0.0(7,16)]triaconta-1(22),7(16),9,13,23,29-hexaene	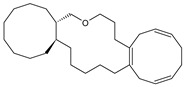	−7.8	27.51–30.19	GLU 166 (1/1)	GLU 166 (6/1), (2.66–3.70) HIS 41 (2/1), (3.74–3.76) CYS 145 (2/1), (3.51 to 3.69)
d	2(3H)-Benzofuranone, 6-ethenylhexahydro-3,6-dimethyl-7-(1-methylethenyl)-, [3S-(3.alpha.,3a.alpha.,6.alpha.,7.beta.,7a.beta.)]-	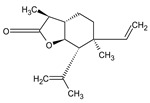	−5.6	0.00–0.00	-	GLU 166 (8/1), (2.80–3.79) HIS 41 (5/1), (3.48–3.96) CYS 145 (1/1), (3.93)
e	Vanillosmin	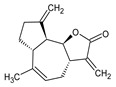	−6.4	30.75–33.33	-	HIS 41 (4/1), (2.65–3.75), CYS 145 (1/1), (3.81)
f	Piperine	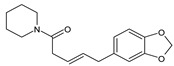	−6.0	30.24–32.01	-	GLU 166 (5/1), (2.15–3.85) CYS 145 (2/1), (3.64–3.82)
g	Stigmasterol	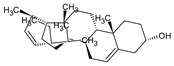	−6.8	31.13–33.56	-	GLU 166 (7/1), (2.57–3.69) HIS 41 (5/1), (3.61–3.89) CYS 145 (3/1), (3.78–3.98)
h	Di(1,2,5-oxadiazolo)[3,4-b:3,4-E]pyrazine, 4,8-diacetyl-	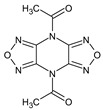	−6.5–−5.7	26.80–33.89	PHE 140 (1/1), HIS 163 (1/1), HIS 164 (1/1)	GLU 166 (14/2), (2.49–3.62) HIS 41 (2/1), (2.96–3.69) CYS 145 (1/1), (3.62)
i	Cycloeucalenol acetate	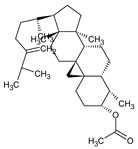	−7.0–6.4	25.29–34.14	THR 26 (1/1) LYS 137 (1/1) MET 276 (1/1)	GLU 166 (8/2), (2.67–3.81) HIS 41 (18/3), (2.87–3.90) CYS 145 (2/2), (3.73–3.77)
k	Strophanthidol	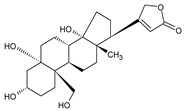	−7.2–6.7	0.00–3.98	THR 26 (1/1) GLU 166 (3/2) GLN 198 (1/1)	GLU 166 (27/3), (1.82–3.69) HIS 41 (8/3), (2.95–3.64) CYS 145 (2/2), (2.94–3.52)
l	Eudesma-5,11(13)-dien-8,12-olide	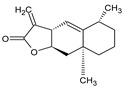	−6.1–5.9	29.56–33.61	-	GLU 166 (2/1), (3.03–3.87) HIS 41(34/2), (1.70–3.96) CIS 145 (3/2), (3.77–3.88)
m	2-Butenoic acid, 2-methyl-, 2-(acetyloxy)-1,1a,2,3,4,6,7,10,11,11a-decahydro-7,10-dihydroxy-1,1,3,6,9-pentamethyl-4a,7a-epoxy-5	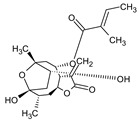	−6.6–6.1	29.19–32.40	(ASN 142)2/2 GLU 166	GLU 166 (6/2), (1.99–3.20) HIS 41(14/2), (1.62–3.78) CIS 145 (6/2), (2.54–3.64)

**Table 2 plants-12-00460-t002:** The zones of inhibition (in mm) of *Saussurea costus* aqueous and acetic acid extracts against examined microorganisms (mean ± standard deviation).

Microorganisms	Acetic Acid Extract (100 mg/mL)	Chloramphgenicol (2.5 mg/mL)	Clotrimazole (5 mg/mL)	DMSO (10% v/v)
*Staphylococcus aureus*	16.5 ± 0.7	26.0 ± 1.4	NA	6.0 ± 0.0
*Bacillus cereus*	25.0 ± 2.8	22.5 ± 0.7	NA	6.0 ± 0.0
*Pseudomonas aeruginosa*	13.5 ± 0.7	15.5 ± 0.7	NA	6.0 ± 0.0
*Escherichia coli*	14.0 ± 1.4	24.5 ± 0.7	NA	6.0 ± 0.0
*Salmonella enterica*	18.0 ± 0.0	23.0 ± 1.4	NA	6.0 ± 0.0
*Candida albicans*	38.5 ± 2.1	NA	16.0 ± 1.4	6.0 ± 0.0

NA—Not applicable; 6.0: the diameter of the paper disk (no inhibition); the unit of the inhibition zone is mm.

**Table 3 plants-12-00460-t003:** The MIC, MBC, MFC, MBC/MIC, and MFC/MIC values of the acetic acid extract of *Saussurea costus* against the examined microorganisms.

Microorganism	MIC (mg/mL)	MBC or MFC (mg/mL)	MBC/MIC or MFC/MIC
*Staphylococcus aureus*	25	25	1
*Bacillus cereus*	25	50	2
*Escherichia coli*	25	50	2
*Salmonella enterica*	25	50	2
*Pseudomonas aeruginosa*	25	50	2
*Candida albicans*	6.25	12.5	2

## Data Availability

The data will be available upon suitable request.

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
