# Peer review of "Inhibitory Activity of *Saussurea costus* Extract against Bacteria, Candida, Herpes, and SARS-CoV-2"

_plants, 2023, doi:10.3390/plants12030460_

Round 1
Reviewer 1 Report (Previous Reviewer 3)
Dear authors,
the overall quality of the manuscript is now improved and most of the methodological inconsistencies have been addressed. Therefore, I think that the manuscript has been significantly improved and now warrants publication in plants
Author Response
Reviewer (1)
The overall quality of the manuscript is now improved and most of the methodological inconsistencies have been addressed. Therefore, I think that the manuscript has been significantly improved and now warrants publication in plants.
Response
Thank you so much for the positive comment, we highly appreciate the time you spent to peer-review our study.
Reviewer 2 Report (Previous Reviewer 4)
The manuscript “Inhibitory Activity of Saussurea costus Extract Against Bacteria Candida, Herpes, and SARS-CoV-2” have been revised substantially. Though the manuscript has some points to address before publication.
- Figure 1 is not at all impressive and informative, better the author can shift the figure as a supplementary file.
- Can the author provide the virtual screening results (Line 339) as a supplementary file? Or at least the author must mention in the text how many compounds were screened for the virtual screening before the selection of these compounds for the docking analysis.
- For the statement “India” line 122, provide the location of the sample collection i.e. city, province, etc., as the phytochemical varied greatly according to the location.
- Line 124, “Dried and powdered samples of … acetic acid solvent (99.8%) (Sigma-Aldrich)” rewrite as “Dried and powdered samples of … acetic acid solvent (99.8%) extraction (Sigma-Aldrich)” followed by filtration and the filtrate was allowed to dry on a glass Petri dish at room temperature. The percentage yield of 50 g extract of air-dried powdered roots was approximately 6%. Delete lines 129-131.
- Line 135 is it a 0.45 mm or 0.45µm syringe filter??? Check it.
- Represent degrees Celsius (lines 139, 141, 142) as oC
1. It would be more informative if an author can provide the IC 50 value for the inhibition study along with the MIC/MBC values.
Author Response
Reviewer (2)
The manuscript “Inhibitory Activity of Saussurea costus Extract Against Bacteria Candida, Herpes, and SARS-CoV-2” have been revised substantially. Though the manuscript has some points to address before publication.
- Figure 1 is not at all impressive and informative, better the author can shift the figure as a supplementary file.
Response
Thank you so much, Figure 1, have been moved to the supplementary files and the figure numbers have been rearranged again
- Can the author provide the virtual screening results (Line 339) as a supplementary file? Or at least the author must mention in the text how many compounds were screened for the virtual screening before the selection of these compounds for the docking analysis.
Response
We have been docked all the identified 69 compounds in order to determine which chemical compounds are better see line 341. Therefore, all screened compounds are mentioned now before the selection of the docking analysis.
- For the statement “India” line 122, provide the location of the sample collection i.e. city, province, etc., as the phytochemical varied greatly according to the location.
Response
Done see line 131
- Line 124, “Dried and powdered samples of … acetic acid solvent (99.8%) (Sigma-Aldrich)” rewrite as “Dried and powdered samples of … acetic acid solvent (99.8%) extraction (Sigma-Aldrich)” followed by filtration and the filtrate was allowed to dry on a glass Petri dish at room temperature. The percentage yield of 50 g extract of air-dried powdered roots was approximately 6%. Delete lines 129-131.
Response
The line 124 have been modified to (Dried and powdered samples of … acetic acid solvent (99.8%) extraction (Sigma-Aldrich)” followed by filtration and the filtrate was allowed to dry on a glass Petri dish at room temperature.) according to your suggestion
Lines 129-131 have been deleted according to your suggestion
- Line 135 is it a 0.45 mm or 0.45µm syringe filter??? Check it.
Response
Line 135 modified from 0.45 mm to 0.45µm
- Represent degrees Celsius (lines 139, 141, 142) as oC
Response
Degrees Celsius modified to oC
- It would be more informative if an author can provide the IC 50 value for the inhibition study along with the MIC/MBC values.
Response
Thank you, in our experience, the IC50 is not employed to assess bacterial cell growth suppression. It might be used to different microbes. The antibacterial agent is often evaluated in a series of 2-fold dilutions since its ability to impede growth is typically only seen over a very small concentration range, typically one or two dilutions. For the IC50 to be properly calculated, a plot of bacterial growth vs chemical concentration is typically excessively steep. The MIC may be used with bacteria as a result. The MIC for bacteria is more of an observation than a figure (as published in various studies!). In a broth microdilution experiment, the lowest component concentration found in the collection of 2-fold serial dilutions at which the bacteria do not reach stationary phase in one day from diluted starting culture is what is seen. We assessed the antibacterial activity of bacteria and yeast in the present investigation. Therefore, it is crucial to standardize the tests between bacteria and yeast, and as you may have observed, they may be conducted in the future for a thorough assessment of this promising plant's anti-Candida characteristics.

Reviewer 3 Report (New Reviewer)
The manuscript is a good work combining docking and experimental approaches to design novel inhibitors for SARS-CoV-2.
Although the work is good but there are some minor modifications needed.
Author Response
Reviewer (3)
The manuscript is a good work combining docking and experimental approaches to design novel inhibitors for SARS-CoV-2.
Thanks for the praise and we appreciate your valuable comments
Although the work is good but there are some minor modifications needed.
- Abstract should have some in silico results not experimental only.
Response
In silico results have been added to the abstract (The binding scores of the investigated molecules were significantly low, ranging from −6.4 to −7.0 kcal/mol. The interaction of sterols with the active site has generally been shown to be more efficient, with beta-Sitosterol showing the most interaction.)
- Line 106-110 missing citations
Reference have been added [ 8]
- The discussion can be improved by adding more references. Some suggested references that can improve the overall discussion are given.
DOI: 10.1002/ptr.6998
DOI: 10.1080/07391102.2020.1776639
DOI: 10.1080/07391102.2020.1769733
DOI: 10.1080/07391102.2020.1779128
Response
The references have been added [ 80,818,82,83]

Reviewer 4 Report (New Reviewer)
Why are yellow marking used in the text.
The scientific soundness of the article is dubious.
Chemical structures of the substances in question should be presented. If you don't know the structue of the pharmacogically active substances, what did the authors use in the molecular docking chapter.
The inhibitory activity of sterols and terpenoids against SARS-CoV-2 was studied using molecular docking. The examined sterols and terpenoids were found to be active - Based on what? This is a simple suposition
Taxonomic and morphologic presentation of the plant is missing.
The quality of the figures is low, and should be improved. I would recommend the use of colour figures, for a better presentation of the results.
The article should be rewritten completly to be published in a journal as Plants
Author Response
Reviewer (4)
Why are yellow marking used in the text?
Response
The yellow markings are all the additions we made in order to improve the manuscript after the first submission and previous modifications and after it has carefully undergone English language editing by MDPI (certificate is attached).
The scientific soundness of the article is dubious.
Response
Respected reviewer,
This comment is completely unacceptable!, Since you did not specify exactly and scientifically the part that led to you claiming that accusation that defames our scientific integrity!
Chemical structures of the substances in question should be presented. If you don't know the structue of the pharmacogically active substances, what did the authors use in the molecular docking chapter.
Response
The Chemical structures of the substances have been added (Table 1)
The inhibitory activity of sterols and terpenoids against SARS-CoV-2 was studied using molecular docking. The examined sterols and terpenoids were found to be active - Based on what? This is a simple suposition
Response
We have been docked all the identified 69 compounds in order to determine which chemical compounds are better (according to your comments we did the docking from the beginning),
Taxonomic and morphologic presentation of the plant is missing.
Response
Taxonomy and morphology of the plant have been added see line 104 -113
The taxonomic details of Saussurea costus are as follows: its Plantae phylum is Trichophyte, its class is Magnoliopsidas, its order is astral, and it belongs to the Asteraceae family. In terms of morphology, Saussurea costus is a 1–2 m tall upright herbaceous plant with a strong and hairy seedhead. The leaves are narrow, rough, glabrous, auriculate, and have irregularly formed teeth in general. The bottom leaves are larger, 0.50 and 1.25 meters in length, and feature long petioles with wings. Upper leaves are tiny and virtually sessile, with two small lobes at the base, which almost wrap around the stem. The flowers are stemless and range in color from bluish-purple to nearly black. Their width varies from 2.4 to 3.9 centimeters, and they are spherical. The Roots are dark brown or gray and reach a length of 40 centimeters.
The quality of the figures is low, and should be improved. I would recommend the use of colour figures, for a better presentation of the results.
Thank you, the quality of the figures have been improved
The article should be rewritten completly to be published in a journal as Plants
Response
The article is rewritten and proofread by professionals from MDPI (Certificate is attached) based on the previous reviewers comments.
We attached the revised manuscript, the supplementary file and the professional English proofread!

Round 2
Reviewer 4 Report (New Reviewer)
Dear Authors
Here are some examples why I have written in the previous review that the article is somehow dubious from a scientific point of view:
All the therapeutic effects mentioned by the authors for different plants are potential effects, since this is not demonstrated by clinical trials. To say that "For example, Allium sativum, Trigonella foenum-graecum, Ferula assa-foetida, Carthamus tinctorius, and Mangifera indica have displayed antidiabetic activity" is misleading. Also "some medicinal plants have exhibited anti-SARS-CoV-2 activity, such as Lycoris radiate, Artemisia annua, Pyrrosia lingua, Nigella sativa, and Houttuynia cordata", really ?
"The highest numbers of cases were reported in the countries that ignored the early warning system and did not respond to the closing of borders" - regarding COVID - this is a supposition not a scientific statement - see what happens now in China
"Several common ailments, including malaria, cholera, and asthma, are treated using medicinal herbs". Medicinal herbs are adjuvant treatments, the protocols to treat this diseases are based on synthetic medicines
Furthermore:
Figure 1 is overshaped, please reduce font size
Verify the chemical denomitaions in Table 1
Figure 4,5 should be in colours not black and white
Check abbreviation not all are defined in the text
Make a list of chemicals used in the experimental part
How did you choose the analytical conditions for the GC method ?
Author Response
Respected reviewer,
Thank you for your careful follow-up and clarifying the reasons for your comment: "somehow dubious from a scientific point of view." Please find below a point-by-point response to your comments. Could you please also check the attached manuscript and check corrections/amendments based on your valuable comments, highlighted in a light-blue color (note: those highlighted in yellow color are for other reviewers).
Here are some examples why I have written in the previous review that the article is somehow dubious from a scientific point of view:
All the therapeutic effects mentioned by the authors for different plants are potential effects, since this is not demonstrated by clinical trials. To say that "For example, Allium sativum, Trigonella foenum-graecum, Ferula assa-foetida, Carthamus tinctorius, and Mangifera indica have displayed antidiabetic activity" is misleading. Also "some medicinal plants have exhibited anti-SARS-CoV-2 activity, such as Lycoris radiate, Artemisia annua, Pyrrosia lingua, Nigella sativa, and Houttuynia cordata", really ?
Thank you for your comments; we clarified that by stating your important notice in the text, and we also added the following statement:
“It is important to mention that, clinical trials are required to demonstrate all these possible activities, particularly regarding SARS-CoV-2, which still no effective drug based on natural products approved yet” (lines 48-58).
Note:
Here, we would like to thank you for this observation because it is consistent with our findings, which found that Saussurea costus showed antiviral activity against the Herpes virus in vitro but did not detect significant activity against SARS-CoV-2, despite claims of the presence of anti-SARS-CoV-2 in many previous published studies, ex.:
https://www.semanticscholar.org/paper/Saussurea-costus-may-help-in-the-treatment-of-Saif-Al-Islam/3b1127fae724e4331bff33f29ed467539330de64
"The highest numbers of cases were reported in the countries that ignored the early warning system and did not respond to the closing of borders" - regarding COVID - this is a supposition not a scientific statement - see what happens now in China
Thank you, you are right! We accordingly deleted this part and replaced it with another quotation from the same reference to show that the applications of some prediction systems may help in control efforts of COVID:
“Electronic health records with easily accessible characteristics in combination with a COVID-19 mortality risk prediction model may allow rapid and precise risk categorization of COVID-19 patients upon admission” Lines 66-68
"Several common ailments, including malaria, cholera, and asthma, are treated using medicinal herbs". Medicinal herbs are adjuvant treatments, the protocols to treat this diseases are based on synthetic medicines
Thank you, your important notice was added to the text (see lines 84-85)
Furthermore:
Figure 1 is overshaped, please reduce font size
Modified
Verify the chemical denomitaions in Table 1
Checked and modified
Figure 4,5 should be in colours not black and white
Done!
Check abbreviation not all are defined in the text
All the abbreviation was defined in the text , thank you
Make a list of chemicals used in the experimental part
Done see 131, All chemicals used (Acetic acid, Dimethyl sulfoxide, ethanol and anhydrous sodium sulfite) were of analytical grade, were used as received without any further purification.
How did you choose the analytical conditions for the GC method ?
See line 144 to 153
By choosing the appropriate column and adjustment of temperature, taking into account the polarity of both the stationary phase and the target analytes. The table below presents the GC analysis conditions
GC analysis conditions see line 144 to 153 and line 154.
The following table is present now in the supplementary file
Table S1. GC-MS analysis conditions
|
Integration |
Gas Chromatography real time analysis software |
|
Detector |
GC/MS QP2012 Ultra |
|
Column |
Rtx-5ms |
|
Column length |
30 m |
|
Inside column diameter |
0.25 mm |
|
Film thickness |
0.25 μm |
|
Column temperature program |
60°C increased at 10°C to 300°C, held for 10 min |
|
Detector temperature |
200 °C |
|
Injector temperature |
250 °C |
|
Carrier gas, inlet pressure |
helium ,1.6ml/min |
|
Split ratio |
0.3 ml/min |
|
Injection volume |
0.1 μl |

Round 3
Reviewer 4 Report (New Reviewer)
The authors improved the quallity of the manuscript and responded to the questions, the article can be published in the current form
This manuscript is a resubmission of an earlier submission. The following is a list of the peer review reports and author responses from that submission.
Round 1
Reviewer 1 Report
In the paper titled “Saussurea costus extract and their inhibition abilities against bacteria, candida, herpes, and SARS-CoV-2,” Idriss et al. characterize the compounds of an extract made from the plant Saussurea costus and present a series of experiments outlining the antimicrobial properties of the plant extract. The question is important. As the authors point out, drug developers are investigating a range of traditional medicines from “Mother Nature,” and understanding the chemical layout of plant extracts is a first step in understanding the possibility of active compounds that may be pharmacologically useful.
While the experimental plan and the overall organization of the manuscript are strong, there are several major aspects of the work that will need to be addressed before it is ready for publication.
Major Comments.
One of the major issues with the manuscript is the language. Although I realize that writing in English when it is not your native language is challenging, the state of the writing right now is not acceptable. The text contains too many language problems to list here, and they range from simple technical items (eg, formatting of organism names, typos, like “molecular mocking”) to more serious grammatical issues like incomplete sentences (line 61) and hard-to-fathom thoughts, including the wording of the title. A complete and thorough editorial review for technical items and English language is needed. And the title definitely needs to be improved.
Regarding the study itself, I have the following major observations.
First, a little more information about the plant sample would be helpful. It was purchased from a an herbalist in Saudi Arabia. Is it possible to know where it was grown? How old was it? How had it been prepared? How did the authors confirm that this was, indeed, the Saussurea costus plant? The Methods say that it was dried—did the researchers dry the root in the lab? If so, how? Were any of the compounds identified in this extract the same as those identified in other studies? I think that this aspect of the manuscript needs to be improved a bit. The section 3.1 that outlines the phytochemical analysis needs to be more targeted toward explaining the actual results of the chemistry as they relate to other studies. In other words, in Section 3.1, the authors nicely outline what others have found with the Saussurea costus plant. But how did the chemistry results here directly compare to other chemical analyses of this plant? In general, the discussion in this section could be improved.
The disc diffusion assays are a little confusing right now. In the Methods section, the authors state that the compounds were dissolved in DMSO and that a DMSO negative control was used. Where are the results from the negative control? The labeling in Figure 5 says “acetic acid extract.” Is that the extract that was put in DMSO? Were there results from the vehicle control discs? That is important to know. And if the compound in acetic acid was added to DMSO, then that is a double vehicle. The negative control would then technically need to be a 1:1 ratio of DMSO to acetic acid as a vehicle control. Also, the table embedded in Figure 5 is not helpful. Most of those statistical metrics are not revealing. Is the last column of that table showing P values? If so, then that is what it should be labeled as. I would totally rethink Figure 5. The images of the zones of inhibition are of low quality, and again, the results of the vehicle control should be included because it may be slightly toxic to some of the microbes. If the vehicle was completely non-toxic for all the microorganisms, that should be mentioned in the text. But most importantly, the authors state that only 2 replicates were done. The disc diffusion assays are not high cost or technically challenging, and a minimum of 3 replicates to more accurately assess variability is the norm. The same goes for the broth dilution assays. This is just sort of a standard of rigor. If it is not possible for the authors to do more replicates of these assays, then this should be noted as a limitation of the study.
The Methods need a bit more information in regard to how the microorganisms were handled. What type of medium were they grown in before inoculation for the disc diffusion assays? What kind of “nutrient” broth was used for the MIC assays? Was it the same medium for all the bacteria? Overall, the Methods would benefit from a little bit more detail and consistency.
Author Response
In the paper titled “Saussurea costus extract and their inhibition abilities against bacteria, candida, herpes, and SARS-CoV-2,” Idriss et al. characterize the compounds of an extract made from the plant Saussurea costus and present a series of experiments outlining the antimicrobial properties of the plant extract. The question is important. As the authors point out, drug developers are investigating a range of traditional medicines from “Mother Nature,” and understanding the chemical layout of plant extracts is a first step in understanding the possibility of active compounds that may be pharmacologically useful.
While the experimental plan and the overall organization of the manuscript are strong, there are several major aspects of the work that will need to be addressed before it is ready for publication.
Major Comments.
One of the major issues with the manuscript is the language. Although I realize that writing in English when it is not your native language is challenging, the state of the writing right now is not acceptable. The text contains too many language problems to list here, and they range from simple technical items (eg, formatting of organism names, typos, like “molecular mocking”) to more serious grammatical issues like incomplete sentences (line 61) and hard-to-fathom thoughts, including the wording of the title. A complete and thorough editorial review for technical items and English language is needed. And the title definitely needs to be improved.
We appreciate the reviewer comment that help us to improve the language style and manuscript
The language has been corrected and the title changed
Regarding the study itself, I have the following major observations.
First, a little more information about the plant sample would be helpful. It was purchased from a an herbalist in Saudi Arabia. Is it possible to know where it was grown? How old was it? How had it been prepared? How did the authors confirm that this was, indeed, the Saussurea costus plant? The Methods say that it was dried—did the researchers dry the root in the lab? If so, how? Were any of the compounds identified in this extract the same as those identified in other studies? I think that this aspect of the manuscript needs to be improved a bit.
Thank you, your important notes were clarified in the following sentence, which was added to the manuscript:
“The root of Saussurea costus of high quality was purchased from “the attar shop” which is a trusted herbal store in Riyadh, Saudi Arabia, with a license from the government to sell herbs; the source of Saussurea costus is India. Notice: (Imported plants and herbs are tested by the government for the quality control including plants types, taxonomy and purity)
The section 3.1 that outlines the phytochemical analysis needs to be more targeted toward explaining the actual results of the chemistry as they relate to other studies. In other words, in Section 3.1, the authors nicely outline what others have found with the Saussurea costus plant. But how did the chemistry results here directly compare to other chemical analyses of this plant? In general, the discussion in this section could be improved.
The section have been modified
The disc diffusion assays are a little confusing right now. In the Methods section, the authors state that the compounds were dissolved in DMSO and that a DMSO negative control was used. Where are the results from the negative control? Thank you, our pre-experimental evaluation on 10% DMSO as a solvent for the dried acetic acid extract showed no activity on the tested microorganisms, supported by many previous publications, it was clarified now and this sentence was added:
“The dried acetic acid extract of Saussurea costus roots was re-constituted by dissolving 500 mg of the dry extract in 10% dimethyl sulfoxide (DMSO) and mixed well using a shaker for up to 1 hour. 10% DMSO has no inhibitory effect on microbial growth, according to our pre-experimental evaluation and literature [39-41]”
The labeling in Figure 5 says “acetic acid extract.” Is that the extract that was put in DMSO? Were there results from the vehicle control discs? That is important to know. And if the compound in acetic acid was added to DMSO, then that is a double vehicle. The negative control would then technically need to be a 1:1 ratio of DMSO to acetic acid as a vehicle control. Thank you, it is a dry acetic acid extract (powder), reconstituted in 10% DMSO (10ml of pure DMSO dissolved in 90ml sterile distilled water) which showed no inhibitory effect on our pre-experimental phase and in many previous publications!
Also, the table embedded in Figure 5 is not helpful. Most of those statistical metrics are not revealing. Is the last column of that table showing P values? If so, then that is what it should be labeled as. I would totally rethink Figure 5. The images of the zones of inhibition are of low quality, and again, the results of the vehicle control should be included because it may be slightly toxic to some of the microbes. If the vehicle was completely non-toxic for all the microorganisms, that should be mentioned in the text. But most importantly, the authors state that only 2 replicates were done. The disc diffusion assays are not high cost or technically challenging, and a minimum of 3 replicates to more accurately assess variability is the norm. The same goes for the broth dilution assays. This is just sort of a standard of rigor. If it is not possible for the authors to do more replicates of these assays, then this should be noted as a limitation of the study.
Thanks. Regarding replications, yes honestly, there were only two replicates. A subtitle "2.11 Research limitations" was written, including this notice.
The image was replaced by a table (Table ……) and the table of the statistical metrics was deleted, and our pre-experimental evaluation of 10%DMSO was added to the table, which was negative.
The Methods need a bit more information in regard to how the microorganisms were handled. What type of medium were they grown in before inoculation for the disc diffusion assays? What kind of “nutrient” broth was used for the MIC assays? Was it the same medium for all the bacteria? Overall, the Methods would benefit from a little bit more detail and consistency. Thank you, type of medium before and after inoculation is mentioned, same for all microorganisms except the yeast. See: Microorganism inoculum was taken from stock culture (in slant tubes containing general-purpose nutrient agar, Oxoid, UK) for bacteria or sabouraud dextrose agar (Oxoid, UK) for the yeast and inoculated in to nutrient broth (general-purpose growth medium, Oxoid, UK) for bacteria or sabouraud dextrose broth for the yeast and incubated for up to 12 hours and up to 24 hours for the yeast to reach the exponential phase.

Reviewer 2 Report
The experimental procedure of extraction with acetic acid and subsequent GC-MS analysis of this extract is not explained. In order to characterize the volatile constituents of the drug, steam distillation of the drug is common.
The characterization of the acetic acid extract is not explained. How were tannins or alkaloids determined? Which substances are these?
The data from Tab. 2 are not supported by experimental data.
Author Response
Reviewer 2
The experimental procedure of extraction with acetic acid and subsequent GC-MS analysis of this extract is not explained. In order to characterize the volatile constituents of the drug, steam distillation of the drug is common.
The characterization of the acetic acid extract is not explained. How were tannins or alkaloids determined? Which substances are these?
The root of Saussurea costus was air-dried at room temperature for many days . The dried roots were ground using an electric grinder. Dried and powdered samples of Saussurea costus were subjected to an acetic acid solvent. Fifty grams of dried Saussurea roots were extracted in 150 milliliters of acetic acid at room temperature for four days with an Orbital Shaker (BioSan PSU-20i). Acetic acid is more ecologically friendly, efficient, and cost-effective than other solvents, making it an ideal extraction technique for phytochemicals, which are valuable constituents in chemical extraction[39]. After filtering the supernatant, the extract was allowed to dry on glass petri dish at room temperature. The percentage yield of 50g extract of air-dried powdered roots was approximately 6%.
The data from Tab. 2 are not supported by experimental data.
The result of the experimental data are presents in SARS-CoV2 entry inhibition of Saussurea costus acetic extract section

Reviewer 3 Report
The manuscript deals with the inhibition capacities (antimicrobial and antiviral) exerted by a medicinal plant raw extract and its chemical characterization by GC-MS analysis. Quality of presentation is very poor and, perhaps for this reason, also significance of content and scientific soundness seem very poor. Major revisions are strictly needed.
-Introduction is only focused on COVID-19 pandemic and antiviral activities of extracts from medicinal plants. Other aspects related to the investigations and the results herein reported must be widely treated and argumented;
-section 2.2 sample preparation and extraction
Why did the authors choose to use acetic acid as a solvent for extraction? This must be argumented;
What about acetic acid solution? Was it an aqueous solution? If yes, which % of acetic acid?
How could the extract be dried on a filter paper? How did the authors recovered the dried residue form the filter paper? Which was the mass of the recovered dried extract?
-section 2.6 Disc-diffusion test
It is not clear which solution the extract was diluted in to perform disc-diffusion test. Was it DMSO???
-section 2.8 MBC or MFC assay
Something is missing in the first sentence. It is nonsense.
-section 2.9.1 citotoxicity assay
It is not clear which solution the extract was diluted in to perform this assay.
-section 2.9.3
It is not clear which solution the extract was diluted in to perform this assay.
-results and discussion
line 236: hydro-distillation extraction?????? How and why was it performed? No description on the M&M section is reported. Please revise
The authors say they have found 109 phytochemical compounds. In my opinion this indication is not properly used. Do fatty acids, carbohydrates and hydrocarbons can be referred to as phytochemicals????
How was it possible to detect, by GC analysis, fatty acids without their prior conversion to methyl esters? How was it possible to detect carbohydrates, by GC analysis, without preliminary derivatization?
-section 3.3 antimicrobial activity
lines 325-326: it is the first time the authors introduce an aqueous extract. How was it obtained? No description on the M&M section is reported. Please revise. Why did the authors choose to compare aqueous and acetic acid extracts?
-figure 5: colors of the column are not distinguishable in this form. Authors should better modify the graph. Histogram columns should be with texturized drawings, not with different colors because in black and white they're not distinguishable.
-section 3.4
Why did the authors compared aqueous/acetic acid extracts for antimicrobial studies and not for antiviral studies? Please be consistent
-Conclusion
line 439: hydrodistillation extract is mentioned again. How and why was it performed? No description on the M&M section is reported. Please revise
Author Response
Reviewer 3
The manuscript deals with the inhibition capacities (antimicrobial and antiviral) exerted by a medicinal plant raw extract and its chemical characterization by GC-MS analysis. Quality of presentation is very poor and, perhaps for this reason, also significance of content and scientific soundness seem very poor. Major revisions are strictly needed. Thank you, the entire manuscript was revised on the light of this recommendation.
-Introduction is only focused on COVID-19 pandemic and antiviral activities of extracts from medicinal plants. Other aspects related to the investigations and the results herein reported must be widely treated and argumented; Thank you, Introduction was enriched according to your recommendation. See: “In the scientific literature, numerous studies on the bioactive properties of medicinal plants have been published. For example, but not limited………..”
“Moreover, some studies reported that Saussurea costus have antiviral activity[9, 10].”
-section 2.2 sample preparation and extraction
Why did the authors choose to use acetic acid as a solvent for extraction? This must be argumented;
Acetic acid is more ecologically friendly, efficient, and cost-effective than other solvents, making it an ideal extraction technique for phytochemicals, which are valuable constituents in chemical extraction
What about acetic acid solution? Was it an aqueous solution? If yes, which % of acetic acid?
The acetic acid solution was aqueous solution %
It was a liquid (99.8%), we mentioned that now: “Saussurea costus were subjected to an acetic acid solvent (99.8%) (Sigma-Aldrich)”
Our bottle!
How could the extract be dried on a filter paper? How did the authors recovered the dried residue form the filter paper? Which was the mass of the recovered dried extract? Thank you for this observation, we followed the mentioned methodology with minor modifications and during the rephrasing of the published methodology we forgot explain all other modified details. Therefore, the entire methodology was revised and mentioned clearly the modification and showed how we used the filter paper disc. See “2.6. Disc-diffusion test”
-section 2.6 Disc-diffusion test
It is not clear which solution the extract was diluted in to perform disc-diffusion test. Was it DMSO??? Thanks a lot, it is now clarified. See “2.6. Disc-diffusion test”
-section 2.8 MBC or MFC assay
Something is missing in the first sentence. It is nonsense. Thank you, it was revised. See : “2.8. MBC or MFC assay”
-section 2.9.1 citotoxicity assay
It is not clear which solution the extract was diluted in to perform this assay.
We appreciate the reviewer's comment allowing us to clarify this topic. A stock of the extract was prepared in DMSO, then the working solution was diluted in DMEM 10% FCS for HEK 293T-ACE2 and in DMEM 2% FCS in the case of Vero cells. To clarify this topic we include the following modification in the text:“A stock solution of 100mg/mL of the extract in DMSO was made. HEK293T-ACE2 or Vero cells were cultured at 1 × 104 cells/well in a 96-well plate in the presence of different concentrations of the S. costus extract in DMEM 10% FBS (Fetal Bovine Serum) for HEK293T-ACE2 cells and in DMEM 2% FBS in the case of Vero cells.”
-section 2.9.3
It is not clear which solution the extract was diluted in to perform this assay.
Thanks to the reviewer to help us to clarify this topic, the experiment was performed in 2.91. the following modification was made in the text.
-results and discussion
line 236: hydro-distillation extraction?????? How and why was it performed? No description on the M&M section is reported. Please revise
Revised to acetic acid extraction (written by mistake)
The authors say they have found 109 phytochemical compounds. In my opinion this indication is not properly used. Do fatty acids, carbohydrates and hydrocarbons can be referred to as phytochemicals????
The phytochemical compounds modified to chemical compounds
How was it possible to detect, by GC analysis, fatty acids without their prior conversion to methyl esters? How was it possible to detect carbohydrates, by GC analysis, without preliminary derivatization?
We appreciate the reviewer comment that help us to clarify this topic we did the measurement without esterification
-section 3.3 antimicrobial activity
lines 325-326: it is the first time the authors introduce an aqueous extract. How was it obtained? No description on the M&M section is reported. Please revise. Why did the authors choose to compare aqueous and acetic acid extracts? Thank you so much for this observation. Aqueous extract is not our target in this paper; it is from our previous study. Being authors from different locations made this confusion. According to your important observation, we deleted any part showing aqueous extract, deleted figure 4, and we transformed figure 5 to a table (Table 3). Thanks a lot.
-figure 5: colors of the column are not distinguishable in this form. Authors should better modify the graph. Histogram columns should be with texturized drawings, not with different colors because in black and white they're not distinguishable. Thank you so much, we replaced the entire graph with a table as a better alternative, clear and concise.
-section 3.4
Why did the authors compared aqueous/acetic acid extracts for antimicrobial studies and not for antiviral studies? Please be consistent. Thank you, I think you mean section 3.3, as section 3.4 did not mention any aqueous extract! In section 3.3, aqueous extract is not our target in this paper; we deleted the aqueous extract and modified the figure to a table and showed also DMSO results which was negative at 10% v/v.
-Conclusion
line 439: hydrodistillation extract is mentioned again. How and why was it performed? No description on the M&M section is reported. Please revise
Revised to acetic acid extract ((written by mistake)

Reviewer 4 Report
1. Line No. 24 Saussurea costus should be written italics
2. Line No. 24 ‘acetic acid extract n extract’. Correct it.
3. Line No. 28 ‘A number of sterols and terpenoids were found to have active properties.’ Which properties author talking about. Complete the sentence with clarity.
4. Line No. 29 ‘Furthermore, the extract exhibited significant antimicrobial activity against bacteria and fungi.’ Specify bacteria and fungi, genus and species
5. Line No. 29-31 ‘The most susceptible microorganism was candida albicans, followed by Bacillus cereus Salmonella enterica, Staphylococcus aureus, Escherichia coli and Pseudomonas aeruginosa, respectively. All the microorganism’s names should be italics.
6. Line No. 33 Correct the word ‘se-lectivity’
7. Line No. 61 ‘Mathematical modeling and data management, sociopolitical, and education.’ This sentence is inappropriate for the context.
8. In the introduction section, it should be clearly defined why author used Saussurea costus for the inhibitory activity against SARS-COVID-19.
9. Line No. 85 ‘All of the reagents were analytical-grade and came from Merck (Germany). ‘Came from Merck’ is a non-scientific language. Correct it according to scientific writing.
10. Section 2.2 (sample preparation and extraction) is ambiguous, It is not clear whether the author is taking filtrate as the extracted materials or the content suspended in filter paper. Explain it. Line No. 89 ‘After filtering the supernatant, the extract was allowed to dry on filter paper’ Is this sentence is correct?? If yes than clarify, how extract can be dried on filter paper?
11. Section 2.3 GC/MS analysis need to explain in more detail with clarity.
12. Add the value of the percentage yield of the extract.
13. Why did the author add anhydrous sodium sulfate to 0.1 g dried extract (line no. 90-91). Usually, sodium sulfate was added to the solution containing some aqueous parts. Make it clear???
14. Explain the process of molecular docking in detail. In the GC/MS section the author obtained the presence of more than 100 compounds. Did the author perform virtual screening before the selection of compounds 105, 104, and 76 for molecular docking? Explain the reason for the selection of only these compounds. The author can refer to this article as a reference: https://doi.org/10.1016/j.ijbiomac.2020.10.135
15. Table 1 is very big and adds no value. It is better if the author can shift this to supplementary material.
16. Is the molecular docking interaction between receptors and ligands only stabilized by the weak vander waal interaction? No hydrogen bonding was observed during the interaction. I suggest if the author may perform a molecular simulation to validate the docking results or autor can provide the distance in Å between the interacting residues of the receptor with a ligand.
17. Add the statistical difference values (ANOVA) in Figures 5, 6, and 8.
Author Response
Reviewer 4
Line No. 24 Saussurea costus should be written italics. Thank you, all scientific names are in Italic now.
- Line No. 24 ‘acetic acid extract n extract’. Correct it.
Corrected to acetic acid extract
- Line No. 28 ‘A number of sterols and terpenoids were found to have active properties.’ Which properties author talking about. Complete the sentence with clarity.
A number of sterols and terpenoids found to have active properties against SARS-CoV inhibition.
- Line No. 29 ‘Furthermore, the extract exhibited significant antimicrobial activity against bacteria and fungi.’ Specify bacteria and fungi, genus and species
Done!
- Line No. 29-31 ‘The most susceptible microorganism was candida albicans, followed by Bacillus cereus Salmonella enterica, Staphylococcus aureus, Escherichia coli and Pseudomonas aeruginosa, respectively. All the microorganism’s names should be italics.
Thank you so much, they are in Italic now.
- Line No. 33 Correct the word ‘se-lectivity’
The word have been corrected to (selectivity) see the text
- Line No. 61 ‘Mathematical modeling and data management, sociopolitical, and education.’ This sentence is inappropriate for the context.
The paragraph have been deleted
- In the introduction section, it should be clearly defined why author used Saussurea costusfor the inhibitory activity against SARS-COVID-19. Thank you, it is mentioned now. See: “Roots of Saussurea costus are widely used in Islamic heritage and famous in Arabia although it is not cultivated there. Moreover, it is claimed between traditional healers that it can control Covid-19 (personal communications)”
- Line No. 85 ‘All of the reagents were analytical-grade and came from Merck (Germany). ‘Came from Merck’ is a non-scientific language. Correct it according to scientific writing.
The paragraph have been modified to (All chemicals used were of analytical grade and were used as received without any further purification and were obtained from Merck (Germany)).
- Section 2.2 (sample preparation and extraction) is ambiguous, It is not clear whether the author is taking filtrate as the extracted materials or the content suspended in filter paper. Explain it. Line No. 89 ‘After filtering the supernatant, the extract was allowed to dry on filter paper’ Is this sentence is correct?? If yes than clarify, how extract can be dried on filter paper?
After filtering the supernatant, the extract was allowed to dry on glass petri dish at room temperature.
- Section 2.3 GC/MS analysis need to explain in more detail with clarity.
The GM/MS from Shimadzu Japanese, with serial number 020525101565SA, and a capillary column (Rtx-5ms-30m0.25 mm0.25 mm), was employed for the samples analysis. The solution of the samples was passed through a 0.45mm syringe filter and into a 1.5mL GC-MS vial, where it was prepared for injection. The specimen was injected utilizing split mode with helium serving as the carrier gas, with a 1.61 ml/min flow rate moving inside the injector. According to the manufacturer, the temperature program started at 50 degrees Celsius at a rate of 10 degrees Celsius per minute and ended at 300 degrees Celsius with a hold duration of 10 minutes. The injection port temperature was 300 degrees Celsius; the ion source was 200 degrees Celsius, and the interface was 250 degrees Celsius. The sample required 35 minutes to analyze in scan mode in the m/z 40-500 charges to ratio range, with a 40-minute run time. The sample's contents were identified by comparing the retention index and mass fragmentation patents of the sample to those in the National Institute of Standards and Technology library. (NIST) [40]. The relative quantities of the different components were determined without correction factors based on the peak area of the GC (FID response).
- Add the value of the percentage yield of the extract.
The percentage yield of 50g extract of air-dried powdered roots was approximately 6%.
- Why did the author add anhydrous sodium sulfate to 0.1 g dried extract (line no. 90-91). Usually, sodium sulfate was added to the solution containing some aqueous parts. Make it clear???
This text has been deleted (anhydrous sodium sulfate to 0.1 g dried extract (written by mistake)
- Explain the process of molecular docking in detail. In the GC/MS section the author obtained the presence of more than 100 compounds. Did the author perform virtual screening before the selection of compounds 105, 104, and 76 for molecular docking? Explain the reason for the selection of only these compounds. The author can refer to this article as a reference: https://doi.org/10.1016/j.ijbiomac.2020.10.135
Terpenoids and steroids were chosen for molecular docking based on virtual screening , reference have been added
- Table 1 is very big and adds no value. It is better if the author can shift this to supplementary material.
Table (1) have been shifted to supplementary material.
- Is the molecular docking interaction between receptors and ligands only stabilized by the weak vander waal interaction? No hydrogen bonding was observed during the interaction. I suggest if the author may perform a molecular simulation to validate the docking results or autor can provide the distance in Å between the interacting residues of the receptor with a ligand.
The distance in Å between the interacting residues have been added see table 2
- Add the statistical difference values (ANOVA) in Figures 5, 6, and 8.
We appreciate the reviewer comment that help us to clarify this topic. After reviewing the manuscript, the figure 5 was removed and included in table 3.
In terms of the figure 6 (new version figure 4), no difference was observed between different SARS Co V2 variants, to clarify this topic the following sentence were included in the text:“No antiviral effect of the acetic extract from Saussurea costus was observed with any SAR-CoV2 variants (Figure 4), also no differences between different variants were observed using ANOVA.”
In terms of the Figure 8 (new version figure 6), the analysis was performed and the statistically significant was included in the new version of the figure, also this text was added: in the figure legend: “Statistical analysis was performed using One-way ANOVA analysis, followed by multiple comparisons with significance indicated as * p < 0.05, ** p < 0.01.”

Round 2
Reviewer 2 Report
Only the volatile ingredients can be characterized by GC-MS. The phytochemical characterization of the acetic acid extract is only possible by HPLC-MS.
The reviewer cannot imagine how tannins or carbohydrates can be determined by GC-MS with the used column and sample preparation.
The concentration used in the disc-diffusion test of 100 mg extract per ml is pharmacologically irrelevant, because this would correspond to a dosage of 100 g per liter.
Without corresponding pharmacological investigations with the indicated substances, the docking studies are not useful.
Reviewer 3 Report
Dear authors,
I've read through the manuscript finding that the overall quality and clarity of presentation has been improved, especially in the materials and methods section. Anyway, I still find some major points to be addressed:
-Introduction section
Even if more references to other medicinal plants and their relative health effects have been added, the main argumented topic is the antiviral activity and the covid-19 pandemic. Other aspects related to the performed investigations (i.e. antimicrobial activity) must be widely treated and argumented;
-Table 3
there's no reference to the unit of measure for the zones of inhibition. Is this cm? mm? Please be consistent in the table and in the footnote.
- GC analysis
It is still not clear how was it possible to detect, by GC analysis, fatty acids without their prior conversion to methyl esters? How was it possible to detect carbohydrates, by GC analysis, without preliminary derivatization?
- Extensive editing of the english language is strictly recommended
Reviewer 4 Report
Authors has been incorporated most of the suggested comments but while reviewing revision a lot of mistakes have been observed related to the grammar, punctuations, references, figure legends, table legends, citation, lack of uniformity. Author advised to address all these issues, according to the following points:
1. Table 1 has been shifted to supplementary material but still legend is there
(Table1: Chemical substance found in Saussurea costus extract via acetic acid), correct it. It should be like Table S1.
2. Table No. should be rectified? If table 1 is now Table S1 than all the table numbers should be changed. Correct it. Table no. 2 is not cited in the text.
3. Where is figure legend of Figure 3?
4. Where is the Figure 4?
5. Line No. 276 check the spilling of Figure 1. Correct it.
6. Figures and tables citation are not proper in the text. All the figures and tables should be properly numbered and should be cited in text. Check this throughout the manuscript.
7. References style is not matching with the journal guidelines, correct it. It is an research article, so references number is too high which can be reduced after removing some old references.
8. Introduction section still need to be improved with clearly mention the objective of study and novelty of study.
9. All manuscript should be checked for punctuation error, especially for space. Author should use uniform style while using % (either 10% or 10 %). Choose one and make it uniform.